# Phage Therapy in Germany—Update 2023

**DOI:** 10.3390/v15020588

**Published:** 2023-02-20

**Authors:** Christian Willy, Joachim J. Bugert, Annika Y. Classen, Li Deng, Anja Düchting, Justus Gross, Jens A. Hammerl, Imke H. E. Korf, Christian Kühn, Simone Lieberknecht-Jouy, Christine Rohde, Markus Rupp, Maria J. G. T. Vehreschild, Kilian Vogele, Sarah Wienecke, Martin Witzenrath, Silvia Würstle, Holger Ziehr, Karin Moelling, Felix Broecker

**Affiliations:** 1Department Trauma & Orthopedic Surgery, Septic & Reconstructive Surgery, Research and Treatment Center Septic Defect Wounds, Federal Armed Forces of Germany, Military Academic Hospital Berlin, Scharnhorststr. 13, 10115 Berlin, Germany; 2Bundeswehr Institute of Microbiology, Neuherbergstr. 11, 80937 Munich, Germany; 3Department I for Internal Medicine, Faculty of Medicine and University Hospital of Cologne, Kerpener Str. 62, 50937 Cologne, Germany; 4German Center for Infection Research (DZIF), Partner Site Bonn-Cologne, 50931 Cologne, Germany; 5Institute of Virology, Helmholtz Centre Munich—German Research Centre for Environmental Health, Ingolstädter Landstr. 1, 85764 Neuherberg, Germany; 6Center for Integrated Infection Prevention, School of Life Sciences, Technical University of Munich, 85354 Freising, Germany; 7Unit for Pharmaceutical Biotechnology, Federal Institute for Drugs and Medical Devices (BfArM), Quality Inspections, 53175 Bonn, Germany; 8Department of General, Visceral, Thoracic, Vascular and Transplant Surgery, University Medical Centre Rostock, 18057 Rostock, Germany; 9Division Diagnostics, Pathogen Characterisation, Parasites in Food, Department Biological Safety, German Federal Institute for Risk Assessment (BfR), 12277 Berlin, Germany; 10Department of Pharmaceutical Biotechnology, Fraunhofer Institute for Toxicology and Experimental Medicine (ITEM), 38124 Braunschweig, Germany; 11Department of Cardiothoracic, Transplantation and Vascular Surgery, Hannover Medical School, 30625 Hannover, Germany; 12Department of Internal Medicine, Infectious Diseases, University Hospital Frankfurt, Goethe University Frankfurt, 60596 Frankfurt am Main, Germany; 13Department Bioresources for Bioeconomy and Health Research, Leibniz Institute, DSMZ-German Collection of Microorganisms and Cell Cultures GmbH, 38124 Braunschweig, Germany; 14Department of Trauma Surgery, University Hospital Regensburg, Franz-Josef-Strauss-Allee 11, 93053 Regensburg, Germany; 15Physics-Department and ZNN, Physics of Synthetic Biological Systems-E14, Technische Universität München, 85748 Garching, Germany; 16Department of Infectious Diseases, Respiratory Medicine and Critical Care, Charité—Universitätsmedizin Berlin, 10117 Berlin, Germany; 17Department of Internal Medicine II, University Hospital Rechts der Isar, School of Medicine, Technical University of Munich, 81675 Munich, Germany; 18Max Planck Institute for Molecular Genetics, Ihnestr. 63-73, 14195 Berlin, Germany; 19Idorsia (Berlin) Pharmaceuticals GmbH, Magnusstr. 11, 12489 Berlin, Germany

**Keywords:** phage therapy, antimicrobial resistance, regulatory framework, Germany

## Abstract

Bacteriophage therapy holds promise in addressing the antibiotic-resistance crisis, globally and in Germany. Here, we provide an overview of the current situation (2023) of applied phage therapy and supporting research in Germany. The authors, an interdisciplinary group working on patient-focused bacteriophage research, addressed phage production, phage banks, susceptibility testing, clinical application, ongoing translational research, the regulatory situation, and the network structure in Germany. They identified critical shortcomings including the lack of clinical trials, a paucity of appropriate regulation and a shortage of phages for clinical use. Phage therapy is currently being applied to a limited number of patients as individual treatment trials. There is presently only one site in Germany for large-scale good-manufacturing-practice (GMP) phage production, and one clinic carrying out permission-free production of medicinal products. Several phage banks exist, but due to varying institutional policies, exchange among them is limited. The number of phage research projects has remarkably increased in recent years, some of which are part of structured networks. There is a demand for the expansion of production capacities with defined quality standards, a structured registry of all treated patients and clear therapeutic guidelines. Furthermore, the medical field is still poorly informed about phage therapy. The current status of non-approval, however, may also be regarded as advantageous, as insufficiently restricted use of phage therapy without adequate scientific evidence for effectiveness and safety must be prevented. In close coordination with the regulatory authorities, it seems sensible to first allow some centers to treat patients following the Belgian model. There is an urgent need for targeted networking and funding, particularly of translational research, to help advance the clinical application of phages.

## 1. Introduction

The decreasing sensitivity of clinically relevant bacteria to antibiotics, as stated in the WHO surveillance reports of recent years, also affects Germany [1,2,3]. The enormous dynamics that can be observed make the rampant spread of multidrug-resistant (MDR) pathogens a medical and health policy challenge of the first order.

The proportion of patients acquiring infections in hospitals is ~3.6%, corresponding to an estimated 400,000–600,000 nosocomial infections in Germany per year [4]. Based on the data of the antibiotic-resistance surveillance of the Robert Koch Institute and the prevalence survey of 2011, an estimated ~6% of these were caused by MDR pathogens, corresponding to 30,000–35,000 infections with MDR pathogens annually. The five most important MDR pathogens alone cause ~29,000 nosocomial infections (~11,000 infections with methicillin-resistant *Staphylococcus aureus* (MRSA), ~4000 with vancomycin-resistant enterococci (*Enterococcus faecalis* and *Enterococcus faecium*), ~8000 with MDR *Escherichia coli*, ~2000 with MDR *Klebsiella pneumoniae* and ~4000 with MDR *Pseudomonas aeruginosa*). Of these, ~1500 cases (0.3%) are caused by isolates that are pandrug-resistant to almost all antibiotic classes [4,5,6,7].

Nosocomial and community-acquired infections by MDR pathogens, according to projections, amount to ~54,500 people in Germany annually. A 2018 study on the disease burden due to MDR pathogens published by the European Centre for Disease Prevention and Control (ECDC) found the number of infection-associated deaths for Germany to be ~2400 people per year (European Union: ~33,000) [8]. The far-reaching health-policy challenges associated with these numbers have been the subject of consultations at all political levels, including the 2015 G7 [9] and 2018 G20 [10] summits, at which further antimicrobial strategies were emphatically called for. Phage therapy has been a particular object of focus in Germany for years now, as evidenced not only by a significant increase in interest from research institutions and some clinics, as well as a substantial increase in patient enquiries, but also by rising interest from policy makers. For example, in 2021, an expert report on the future significance of phage therapy for Germany was commissioned by the German Bundestag (Health Committee and Committee for Education, Research and Technology Assessment) and is expected to be published in 2023. Against this background, the current situation of phage therapy in Germany will be presented here.

## 2. Methods

The current situation of phage therapy in Germany is presented primarily from the perspective of clinicians, cooperating research institutions, and regulatory authorities (BfArM, Federal Institute for Drugs and Medical Devices, Bonn, Germany), with whom effective exchange on the topic has been in place since 2017. Thus, an interdisciplinary network of physicians, pharmacists, microbiologists, bioengineers, bioinformaticians and regulators—all with a patient-oriented focus—were involved in describing the current situation in Germany. The goal of taking translational aspects into account in the present analysis seemed achievable within this group of authors, following scientific exchange at national and international meetings over the last five years, as well as joint research activities. An attempt was made to include only one author from each working group. The following aspects are addressed in particular:The regulatory situation or legal framework for phage products in Germany;The extent of the existing clinical application of phages;Existing activities to produce phages for this purpose;The status of phage banks;The technical status of sensitivity testing (“phagogram”).

For an insight into further developments, ongoing research projects, the activities for a one-health approach, as well as the existing Germany-wide network structure and activity will be presented.

To avoid a limited overview of the phage landscape in Germany, literature covering at least the years 2020–2022 was reviewed for the academic and industrial working groups that, according to the authors, have contributed to translational phage research. In addition, the authors of this article provided their opinion on the most urgent changes needed in the German-phage landscape, the most important research projects, and the biggest hurdles for the implementation of phage therapy. Multiple answers were possible.

## 3. Results

### 3.1. Current Regulatory Situation

Within the European Union, phages for therapeutic use in humans are defined as biological medicinal products, in accordance with Directive 2001/83/EC. This implies the need for a marketing authorization (MA) for market access of phage-therapy medicinal products (PTMP). Article 3 of Directive 2001/83/EC (and paragraph (§) 21(2b) of the German Medicinal Products Act, ‘AMG’) define exemptions from the need for an MA, including prescription medicinal products prepared in a pharmacy for a specific patient (“magistral formula”). According to §55(8) AMG, these products must be manufactured in compliance with the recognized pharmaceutical rules (e.g., European Pharmacopoeia, Ph. Eur.). A Ph. Eur. general chapter on PTMPs is currently being drafted that will provide harmonized quality standards. According to §13 AMG, a manufacturing authorization is required for PTMPs produced on a commercial/professional basis and intended to be marketed. An exemption exists (§13(2) AMG) which includes manufacturing in pharmacies (“within the scope of normal pharmacy operation”) or by physicians (“manufactured under their direct professional responsibility for the purpose of personal use on a specific patient”). PTMP manufacturing must also follow the recognized pharmaceutical rules, and is subject to notification of competent supervisory authorities (§67(2) AMG). PTMPs therefore can be made available to patients using either way, MA (and clinical trials) or formula magistralis.

While MA would primarily pertain to standardized ready-to-use PTMPs, magistral phage products could provide a more personalized approach, adapted individually. The existing regulations are often seen as an obstacle for phage therapy. Nevertheless, as indicated above, varying routes for patient access to PTMPs are currently available. In any case, the development and modification of regulatory requirements will be based on the expansion of knowledge. Once the efficacy of PTMPs is demonstrated, the regulatory framework can be amended, if required, to address the particularities of these products. However, they can be used in cases of hardship. Thus, within the framework of an official compassionate-use program, the use of PTMPs that are currently included in clinical trials or in the approval process of an MA application can be authorized by the national authorities, and thus would be available for a defined group of patients (Table 1).

This officially authorized compassionate-use program is distinguished from an “individual treatment trial”, which does not require notification. The latter is the application of PTMPs in an individual case, decided on by the treating physician, under his or her responsibility, within the framework of therapeutic freedom and with the patient’s consent. The individual treatment is usually applied after all standard treatment options have been exhausted or are not available, i.e., in the case of an “unmet medical need”, according to Article 37 (i.e., unproven interventions in clinical practice) of the Declaration of Helsinki of the World Medical Association (WMA), and if the treating physician suspects a benefit of phage use for the patient, based on scientific findings. The focus of the individual treatment trial is not on obtaining research results, but on healing the individual patient. However, the regulatory authorities explicitly emphasize that precise documentation, observation of the patient and transparent communication of the individual case, as well as scientific justification, must be in place, which also corresponds to the requirement of Article 37, Declaration of Helsinki. No therapeutic phage products are currently clinically approved in Germany; an overview of ongoing and planned clinical trials is included in Section 3.6, below.

### 3.2. Clinical Application of Phages

#### 3.2.1. Brief Overview of Phage Use in Germany from the 1930s to Modern Times

At the beginning of the Second World War, phage therapy experienced a significant upswing in Germany. Dysentery-Polyfagin^®^ against *Shigella* infections was the first phage preparation marketed by Behringwerke, near Marburg/Lahn, in July 1939 [11], followed shortly thereafter by Typhoid-Polyfagin^®^ against *Salmonella*. Many German soldiers were subsequently treated with phages. Positive therapy results were reported, but also—after the initial euphoria—relevant failures. It has been speculated that many of these failures occurred because clinicians were not aware of the narrow host specificity of the phages and important application parameters, and, probably, temperate phages were also used. In addition, process-related impurities, and insufficient stability of the phage solutions impaired product quality. However, scientific proof for effectiveness as well as for the manifold speculative reasons for failure is lacking. Ultimately, these failures greatly reduced interest in phage therapy after the market launch of antibiotics (penicillin and sulfonamides) in the clinical field [11].

In the German Democratic Republic (GDR), a ready-to-use phage preparation, Intestolysin^®^, was available from 1959 to 1965 (against typhoid, paratyphoid, dysentery, and pathogenic *E. coli*, produced by the Institute for Disease Control in Berlin-Weißensee). This preparation was, however, ineffective against *Shigella flexneri* type 4A, which was responsible for an epidemic in 1962. Within a very short time, production capacities of 11,000 L per week were implemented for the prophylaxis of the population (administration: duration, 10 days, total, 50 mL, titer, up to 10^12^ plaque-forming units (PFU)/mL). At that time, phages in the GDR were not subject to the *“Ordinance on the Regulation and Supervision of the Traffic in Medicinal Products”, but to the “Ordinance on the Traffic in Vaccines, Sera and Bacteriophages”.* Phage prophylaxis began in Berlin on 26 April 1962. Until June 14, 1962, a total of 174,906 people received the phage preparation [12]. Efficacy data of the prophylactic measure is not available; however, the effort was discontinued on 19 June 1962, suggesting that the epidemic was likely considered overcome.

Since August 1979, phages have been used to treat bacterial infections in the Endo-Klinik Hamburg (H. Lodenkämper), over a period of at least 5 years (unpublished data). For the treatment of periprosthetic infections, the phage lysate was administered to the patient twice daily for 14 days before surgery and for 14 days afterwards (10 mL phage broth, 10^7^–10^8^ PFU/mL; 30 min before eating; after taking an acid-binding drug). In addition, the phage lysate was introduced intraoperatively into the infected area (exposure time 30 min). From August 1979 to May 1981, 54 patients with implant-associated infections were treated (Figure 1).

Around 1995, phages (usually from Georgia) began to be applied topically to chronic limb wounds, post-traumatic infections and burn wounds at the Hanover Medical School (B. Wippermann), as part of individual treatment trials (unpublished data). Most recently, this treatment series was continued in Hildesheim. Since 2015, 33 patients with chronic periprosthetic infections (on average 11 previous operations) have been treated topically with phages by another working group (R. Ascherl) in the Chemnitz and Tirschenreuth hospitals and, with a follow-up period of 14–42 months at that time, a complete cure of infection without severe side effects (mostly fever and chills on the 2nd–4th day) was achieved in 25 cases. (In twenty-three patients, antibiotics were used simultaneously, in two they were used subsequently to the treatment).

#### 3.2.2. Phages Use in Germany Today (without Exception under the Umbrella of Article 37 of the Declaration of Helsinki), with a Brief Outlook for the Coming Years

Berlin: At the Military Academic Hospital Berlin, three patients have been treated with phages from Georgia since 2016, partly in collaboration with the laboratory for molecular and cellular technology (LabMCT) at Queen Astrid Military Hospital in Brussels (Belgium) and the Charité—Universitätsmedizin Berlin, Germany, Center for Musculoskeletal Surgery. As part of the *PhagoFlow* research project (see below), it will be possible to treat patients with *P. aeruginosa* and *S. aureus* infections on a larger scale with magistral phage preparations, from spring 2023 onwards (www.phagoflow.de). Additionally, at the German Heart Institute Berlin, Germany, six patients were treated as a last-resort measure between December 2018 and October 2021, in collaboration with Charité—Universitätsmedizin Berlin, Germany, Center for Musculoskeletal Surgery, with phages acquired from the Belgian LabMCT and Eliava Institute of Bacteriophage, Microbiology and Virology, IBMV (Tbilisi, Georgia). Following skin closure, a 1:1 mixture of PYO-bacteriophage with 10^6^ PFU/mL and staphylococcal phage Sb-1 with 10^7^ PFU/mL (5 mL, every 8 h for 10 days) was applied locally—and in one case also orally and intravenously [13,14]. Out of three patients treated locally with commercial phage preparations, two (infected with *S. aureus*) remained infection-free until the end of the observation period (≥30 months) [14]. Furthermore, one patient with a *P. aeruginosa* infection, followed by *Staphylococcus haemolyticus*, received a magistral phage preparation applied both locally and intravenously, and remained infection-free for four months before succumbing to LVAD pump thrombosis. Another patient, with an *S. aureus* infection, was administered PYO-bacteriophage plus Sb-1 locally and, at a nine-month follow-up, showed no signs of a local infection [13].

Hannover: Nowadays, phage therapy is again based at the Department of Cardiac, Thoracic, Transplantation and Vascular Surgery of the Hannover Medical School. The department’s portfolio includes 33 cases of personalized phage therapy since 2015 in critically ill patients, of which 31 were successful. The high clinical success rate (>90%) is due to a combination of modern principles of permission-free preparation using mostly self-isolated phages (see below), an interdisciplinary approach to administration of phages and an optimal, concomitant conventional treatment, including antibiotic therapy [15]. In the setting of an academic clinical center, the selection of well-characterized phages using established phagogram methods, the on-demand production of selected phages in the department’s laboratories, and application in the most rational manner are ensured. Among all treated cases the largest cohort were patients with implant-associated infections, including infections of vascular prostheses (n = 7), left ventricular assist devices (LVAD; n = 4), pacemakers/pumps (n = 3) and surgical-site wounds (n = 8). These were followed by respiratory-tract infections, including ventilator-associated pneumonia and pleural empyema (n = 4), as well as transplant-associated infections under drug-induced immune suppression (n = 4). Among the bacteria (including those as part of a mixed infection), subject to phage therapy were *S. aureus* (n = 13), *P. aeruginosa* (n = 12), *E. coli* (n = 4), *E. faecium* (n = 3), *K. pneumoniae* (n = 3), *Burkholderia multivorans* (n = 2), *E. faecalis* (n = 1), *Stenotrophomonas maltophilia* (n = 1), *Serratia marcescens* (n = 1), and *Proteus* spp. (n = 1).

Regensburg: At the Department of Trauma Surgery University Hospital Regensburg, phage therapy has been performed since October 2022. The first case was an infected non-union of the proximal femur with different MDR gram-negative bacteria. Phage testing was performed by LabMCT at Queen Astrid Military Hospital and treatment with Intesti^®^ bacteriophage cocktail produced by the Eliava Institute was recommended, which is active against various strains of *Shigella*, *Salmonella*, *E. coli*, *Proteus*, *S. aureus*, *P. aeruginosa*, and *Enterococcus*. To improve local release kinetics, the phage cocktail was mixed with a hydrogel, as previously described [16]. No local or systemic side effects have been observed after a three-month follow-up. Additionally, the department will participate in the *PhagoDAIR* trial, a randomized, non-comparative, double-blinded phase I/II clinical study in patients with *S. aureus* knee or hip periprosthetic-joint infections with indication for debridement, antibiotic and implant retention (DAIR), combined with antibiotics (ClinicalTrials.gov ID: NCT05369104).

Rostock: Phage therapy is also performed at the Clinic for General, Visceral, Thoracic, Vascular and Transplant Surgery, Rostock University Medical Centre. In the field of vascular surgery, it is mainly patch- or bypass-associated infections in the groin area, followed by acute life-threatening aortic-prosthesis infections. In cardiac surgery, the most frequent indication is drive-line infection after LVAD insertion. In orthopedic-surgery patients, periprosthetic-joint infections in the hip area dominate, increasingly also after implantations of endo-exo systems in transfemoral amputations. Unfortunately, despite a high demand for treatment, there are fundamental problems with individualized phage therapy. Therefore, it has only been possible to treat patients with composite phage cocktails (SniPha 360, Phage 24, Tyrol, Austria), which have so far been accompanied by therapeutic success of only ~30%. The focus on individualized phage therapy with on-site production planned for 2023 is expected to significantly improve success rates. Current and planned research projects are focusing on testing the thermostability of phages in bone cement, topical phage application to infected endo-exo systems at the stoma, and the planning of the study “Bacteriophage-related treatment of chronically infected Drive-Line/LVAD Systems: a randomised controlled multicentre trial” (via a German Research Foundation (DFG) application), in collaboration with the Heart Center, Leipzig, Germany.

### 3.3. Production of Phages

The Fraunhofer Institute for Toxicology and Experimental Medicine (ITEM), Braunschweig, is developing a platform-like manufacturing process for natural phages as APIs using host bacteria under GMP-conforming conditions. For this purpose, the starting materials, master cell banks (MCB) and master phage stocks (MPS), are first produced according to GMP. Phage drug production begins with cultivation (up to 10 L) of the production strain from the MCB, infection with phages (MPS) and subsequent cell lysis. The phage-containing lysate is first separated by depth filtration (to reduce cell fragments), diluted in a buffer suitable for chromatography (diafiltration) and then purified chromatographically to remove endotoxins and host-cell proteins. This is followed by a second diafiltration step with a physiological buffer and 0.2 µM filtration, to reduce bioburden. Finally, the phage agent is filled into the primary packaging. The challenge in developing the manufacturing process is to find suitable conditions that allow for high titers and yields and the reduction of endotoxins and host-cell proteins. After filling, the phages must meet international quality requirements and are tested for identity (sequencing), potency (plaque assay) and purity (e.g., endotoxins), for release. The justification of the batch-release specifications is carried out in close exchange with the competent authorities and process-immanent aspects. The challenge here is that phages represent a completely new class of active substances, for which official quality standards remain to be defined. The first manufacturing authorization for phage production in Germany was granted in August 2022. Three phages produced this way are set to be supplied to the Military Academic Hospital Berlin in Q1 2023.

An alternative way to produce phages, the expression in an in vitro system, is currently being developed by INVITRIS (https://invitris.com, accessed on 12 January 2023), a spin-off company of the Technical University of Munich. For in vitro or cell-free protein synthesis (CFPS) the cellular-expression machinery is isolated from the cell, most commonly *E. coli*, and combined with metabolites, precursor molecules and a DNA template specific for the intended phage. CFPS has been used to produce *E. coli* phages such as T7, phiX174, and MS2 with titers of up to 10^12^–10^13^ PFU/mL [17]. Recent improvements to the system enabled the expression of non-*E.-coli* phages, including those against clinically relevant pathogens such as *Yersinia pestis* and even gram-positive bacteria such as *Bacillus subtilis* [18]. Production of therapeutic agents in CFPS reduces endotoxin levels and the use of a well-characterized donor strain eliminates concerns of prophage contamination. CFPS is easily scaled to different volumes, and the host-independent phage production in a single system allows for personalized applications. In vitro phage production is also a promising approach for phage engineering of the genome and the capsid [18,19]. A test case showed the personalized application of phages against a carbapenem-resistant *K. pneumoniae* patient isolate, from isolation to in vitro production. Utilizing the advantages of in vitro bacteriophage production, a personalized phage-therapy pipeline can be implemented, and could be applied for on-site production [20,21]. Such a personalized approach is especially important for pathogens with narrow specific phages, such as *Klebsiella* phages, most of which initiate infection through the highly diverse capsular polysaccharide of *K. pneumoniae*.

A production process according to §13(2b) AMG (permission-free production of medicinal products) that uses host bacteria is currently only in use at the Hanover Medical School (Clinic for Cardiac, Thoracic, Transplantation and Vascular Surgery) according to the quality and safety requirements for sustainable-phage-therapy products. The production process uses only strictly lytic phages, which do not contain known genes encoding integrases, bacterial virulence and antibiotic-resistance factors. Despite the absence of a conventional GMP process in the production consumables, nutrient media and reagents designed for either GMP production or for clinical use are employed. For example, nutrient media free of animal components are used for cultivation. The host bacterial strains are tested for absence of inducible prophages. Phage amplification is performed on a small-scale, using dense nutrient media, which has shown stable high-phage titers and does not require optimization of the production process. After amplification, the phage lysate undergoes a multistep purification process to remove unlysed cells, nutrient-medium components, and pyrogens, followed by control of sterility, phage titer, identity, and endotoxin content. Other clinics in Germany (Berlin, Regensburg, Rostock) are currently also aiming for this type of production.

### 3.4. Status of Phage Banks

In the early 2000s, a working group specializing in clinically relevant bacteria was founded at the Leibniz Institute DSMZ and, furthermore at the DZIF (German Center for Infection Research) strain repository. Both provide an essential basis for isolating, characterizing, and investigating phages at the DSMZ and for using them in the institute’s so-far exclusive applicative-phage-research projects. The bank contains >1000 phages for more than 150 bacterial species; however, phages for the ESKAPE bacteria (*E. faecium*, *S. aureus*, *K. pneumoniae*, *A. baumannii*, *P. aeruginosa* and *Enterobacter* spp.) are dominate, and are used in externally funded projects such as *P. aeruginosa* in *Phage4Cure*, some prioritized species in *PhagoFlow*, *E. coli* in *IDEAL-EC* and *E. faecium* in *EVREA-Phage* (there are 30–150 phages per ESKAPE species). Other large phage panels exist for *Achromobacter xylosoxidans* and *Stenotrophomonas maltophilia*; phages in lower numbers were isolated on rarer patient isolates of, e.g., *Bordetella bronchiseptica*, *Burkholderia cepacia* or *Mycobacterium abscessus*. The focus of the DSMZ phage bank is defined by medical need. DSMZ can search for new suitable phages at any time, at the request of clinicians, and they can then be supplied according to material-transfer agreements. However, direct application in humans is excluded, for warranty reasons. For genomic-identity confirmation, the DSMZ was granted GMP certification according to §64 (3f), German Medicinal Products Act, in 2022.

Another large phage bank is located at the Bundeswehr Institute of Microbiology (IMB), Munich, with a focus on the isolation and characterization of *Klebsiella* phages specific for highly resistant clinical isolates of 3 to 4 MRGN *K. pneumoniae* (i.e., resistant to 3-4 of 4 important antibiotic groups) [22]. The IMB phage group has one of the most extensive collections of 3-4 MRGN *K. pneumoniae* (>380 isolates with a large variety of capsule types (>80)) and their specific phages (>505 plaque-purified stocks) in Europe. The group is using its phage collection for studies to extend the host range [23], and engineers reporter phages to monitor *K. pneumoniae* infections [24].

Phage research at the German Federal Institute for Risk Assessment (BfR) is based on fundamental investigation of the occurrence of phages (temperate/lytic), their genetic diversity and their potential for application, especially in the veterinary and food sectors. Therefore, the culture collections of foodborne pathogens as well as bacteria recovered from livestock during annual monitoring of zoonoses are used for phage testing. Phages against foodborne pathogens at the BfR mainly originated from samples of the food-production chain (i.e., process water of slaughterhouses, food products, livestock), but have also been recovered from environmental sources (i.e., wildlife) and/or municipal/clinical wastewater. The current collection includes phages against *Salmonella* (>120 phages), *K. pneumoniae* (>80), *Yersinia* (>50), *Brucella* (>40), *P. aeruginosa* (>30), *Morganella* (>15), *Campylobacter* (>15), *Burkholderia* (>15) and *Vibrio* (>10). However, due to the different clonality of the bacteria in the human and livestock/food sector, most of the phages are much more effective on isolates of the respective compartments than on human isolates (esp. *Klebsiella*).

### 3.5. Status of Susceptibility Testing (Phagogram)

Conventionally, phage–host-susceptibility testing is based on the double-agar-overlay plaque assay [25], which is also used to determine the concentration of infectious phage particles in a sample. For this purpose, an agar plate is overlaid with a soft agar containing a culture of the bacterial isolate and a dilution of the phage sample to be examined. After incubation, plaques (clear zones in the bacterial lawn) can be counted at suitable dilutions. A resource-saving modification of the method is the spot assay, in which phage dilutions are not added to the soft agar, but spotted onto it. Analogous to these methods, susceptibility testing of a patient isolate to available phage substances can be performed. Plaque formation indicates phage susceptibility to the bacterium. In parallel, the concentration of the phage on its production hosts is determined, to obtain the efficiency of plating (EOP). In general, the higher the titer of the phage on the patient isolate compared to the production host, the higher the expected potency for the patient. The disadvantage of these methods is the low automatability and thus the necessity of manual steps. In contrast, susceptibility testing in the planktonic-killing assay is performed in a 96-well plate format, in which liquid culture of the patient isolate is infected with phage lysates [26]. Incubation takes place in a plate reader that records the optical density of the culture. The faster and more completely the culture is lysed at low phage load, the more likely treatment success. The advantages of the planktonic-killing assay are low material and time requirements, the possible automation with pipetting robots and the possibility of testing the effect of phage cocktails and supplementation of antibiotics.

### 3.6. Ongoing Translational Research Topics and Projects and Clinical Trials

“Phage4Cure” (duration 2017–2024, lead M.W., Berlin-Braunschweig, Germany, https://phage4cure.de, accessed on 12 January 2023, funded by the German Federal Ministry of Education and Research (BMBF)) is a first-in-human (FIH) study to investigate safety, tolerability, and preliminary efficacy of a phage cocktail in healthy volunteers and patients with chronic *P. aeruginosa* lung infection. The phages investigated in the clinical trial were selected with the help of the phage biobank of the Leibniz Institute DSMZ GmbH. Three phages from the order of *Caudovirales* were selected for the study: two myoviruses (JG005 and JG024), and one podovirus (Bhz17). JG005 and JG024 had been known previously, whereas Bhz17 was newly isolated. The phage selection aimed at providing a broad antibacterial spectrum and limiting the emergence of phage-resistant bacterial variants. Three individual IMP formulations of the respective phages were initially prepared, according to European GMP standards and will subsequently be combined in a cocktail prior to use. The cocktail will be administered as an aerosol through a CE-certified nebulizer. Efficacy and safety were previously tested, in animal models. The *Phage4Cure* study consists of two parts: part 1 is a classical single-ascending-dose (SAD) design in healthy volunteers, while part 2 is a multiple-dose design conducted in patients with chronic *P. aeruginosa* colonization of the lung, with demonstrated susceptibility to this particular phage combination. All documentation for phases 1a (phage inhalation by healthy individuals) and 1b (initial efficacy demonstrations in a small group of patients with bronchiectasis and *P. aeruginosa* colonization) has been completed, allowing for phase 1a to start in Q2 2023. In the longer term, a scalable procedure for phage therapy will be developed, which, after initial establishment, can also be applied to other phage entities in its modified and adapted form.

“PhagoFlow” (duration 4/2019–3/2024, lead C.W., Berlin-Braunschweig, Germany, https://www.phagoflow.de/en/, accessed on 1 January 2020; funded by Innovation Fund, Federal Joint Committee, G-BA). The project investigates whether, with today’s biopharmaceutical possibilities, it is possible to adapt phage preparations in the hospital pharmacy to the individual patient and prepare them in time for therapeutic use—thus, the practicability of magistral production of phage products is investigated (rather than a clinical study in the real sense). The project focuses on wounds on arms and legs infected by multidrug-resistant pathogens. In the first project phase, phages were isolated (DSMZ GmbH), characterized and preserved with all methodology following the OECD Best Practice Guidelines for BRCs. Subsequently, they will be produced in a biotechnological process (Fraunhofer ITEM) in such a way that they can be made available in a purified form to the hospital pharmacy, as API components. The second phase of the project is aimed at treating patients. First, pathogens from a patient’s wound material are typified for phage sensitivity and then a tailor-made phage preparation is produced. Treatment, initially with phages against multidrug-resistant *P. aeruginosa*, will begin in February 2023, and against *S. aureus* from August 2023.

“MAPVAP” (pre-clinical mechanistic assessment of two bacteriophage cocktails targeting multidrug-resistant Pseudomonas aeruginosa and Escherichia coli for the treatment of ventilator-associated pneumonia; duration 1/2020–6/2023; co-lead: M.W., Berlin). The MAPVAP project is a collaborative research program financed by France and Germany through the 2019 dedicated call on antimicrobial resistance (BMBF & Agence Nationale de la Recherche (ANR, France)) [27]. Ventilator-associated pneumonia (VAP) is frequently caused by MDR bacteria, with P. aeruginosa and Enterobacteriaceae being most relevant. Using two established bacteriophage cocktails specific for P. aeruginosa or *E. coli* that originate from German and French teams, respectively, MAPVAP joins forces to (1) characterize phage-resistance development during in vivo preclinical treatment, (2) decipher the impact of these cocktails on the respiratory and intestinal microbiota, as well as on microbiota-dependent immune responses, (3) evaluate the efficacy of the cocktails in penetrating biofilms produced in vitro and ex vivo on explanted human-lung tissue, (4) investigate the direct interaction of phages with the immune system, and the mechanistic basis for the synergy between innate immune cells and phages during therapy, and (5) characterize by mathematical modeling the efficacy of these cocktails in vivo, and propose optimized treatment regimens [27].

“IDEAL-EC” (duration 2021–2022, lead A.Y.C., Cologne). In collaboration with the Leibniz Institute DSMZ and DZIF, the *IDEAL-EC* project focused on evaluating phages against extended-spectrum beta-lactamase-producing clinical *E. coli* samples. The WHO has recently classified *E. coli* as a pathogen of international concern. Antibiotic-resistant *E. coli* strains are on the rise, and can cause severe infections, especially in immunocompromised patients. During the project period, an *E. coli* phage cocktail was developed, which is currently being investigated in preclinical models (in vitro intestinal model, mouse model). The results of this project will serve as the basis for subsequent preclinical and clinical studies investigating the effects of *E. coli* phages on the human microbiota and their applicability for therapeutic purposes.

“EVREA-Phage” (duration 7/2022–7/2025, leads C.R. and Johannes Wittmann, Braunschweig). The DZIF-funded, translational preclinical project *EVREA-Phage* envisages the characterization of new phages and the composing of a phage cocktail against *E. faecium* for oral application, to specifically decolonize intestinal VRE *E. faecium* in immunocompromised patients. Considering potential concomitant colonization by different *E. faecium* strains in individual patients involving not only VRE but also VSE (vancomycin-sensitive) or VRE+ strains being additionally resistant to linezolid, the phages selected for the cocktail will be tested against panels of all mentioned *E. faecium* variants, to ensure the broadest-possible host coverage. Large phage and strain panels were compiled to perform the full biological-phage characterization. Bacterial strains are clinical isolates from German university hospitals of the DZIF network and from the Robert Koch Institute, and all phages were newly isolated by the Leibniz Institute DSMZ. *EVREA-Phage* envisages the demonstration of phage efficiency by in vitro and in vivo gut models at the University Hospital Bonn and the Helmholtz Centre for Infection Research, Braunschweig, to confirm phage effects, according to regulatory requirements and the BfArM scientific advice. All derived data of *EVREA-Phage* will flow into the preparation of an IMPD (investigational medicinal product dossier) required for later authorization of a clinical trial, applying highly purified approved phage preparations.

“Phage2030” (duration 2022–2023, leads C.W. and F.B., coach K.M., Berlin) aims to identify feasible research efforts and manufacturing pathways, the coordination and targeted support of which would represent a “leap forward” in making phage therapy available to the German public by 2030. The validation study required for this is funded by the Federal Agency for Disruptive Innovation, SPRIN-D (www.sprind.org). The project is currently identifying top groups of applied research in Europe to form clusters of excellence and to focus efforts, for example, on innovative and rapid approaches to match patient isolates to phages, on improving the yield of phage-production processes and on exploiting synergies between phages and antibiotics. In addition, with a focus on ESKAPE pathogens, requirements for large-scale production of phage cocktails for bacteria with broad-spectrum phages (e.g., *S. aureus*, *P. aeruginosa*) or small-scale on-site production for individualized phage therapy for pathogens with narrow-spectrum phages (e.g., *K. pneumoniae*, *A. baumannii*) are being evaluated.

“CRAB-A-Phage” (duration 2023–2025, lead A.Y.C., Cologne). Another project with close cooperation between the Leibniz Institute DSMZ and the DZIF will focus on the pre-clinical evaluation of phage–antibiotic synergy in multidrug-resistant *A. baumannii*. Antibiotic-resistant *A. baumannii* is among the six leading pathogens, accounting for most of the deaths associated with antibiotic resistance, while new antimicrobial treatment options are scarce. Results of *CRAB-A-Phage* will not only provide important insights into phage–antibiotic interactions, but also into the role of potentially evolving phage resistance. It is intended to transfer the results into the clinical setting and improve current experimental treatment approaches with phages against *A. baumannii*.

Helmholtz Centre Munich and Technical University of Munich (head L.D.) Conducting multiple projects funded by the European Research Council Innovative Training Networks (ITN; ‘VIROINF’, duration 2020–2024), DFG priority program ‘Novel Concepts in Phage Biology’ (duration 2021–2024) and sequencing program ‘Asthma-Phage’(2023–2025) that aim to explore the cross-talk between virome, microbiome and the human host in dysbiosis-associated digestive and respiratory diseases (e.g., asthma) and to eradicate causative bacteria using synthesized-phage communities (phageome therapy). To this end, multiple tools have been developed, including Replidec to predict phage replication cycles [28] and Viroprofiler to characterize uncultured phage communities (https://github.com/deng-lab/viroprofiler/, accessed on 12 January 2023”) [29,30]. The projects, led by L.D., also include “Phage therapy Against Colorectal-Cancer-Associated Helicobacter and NEC Pathogens” (duration: 2019–2026, funded by DFG). In cooperation with the Collaborative Research Centre ‘Microbiome Signatures—Functional Relevance in the Digestive Tract’, the aim is to eradicate pathobionts linked to colorectal cancer and necrotizing enterocolitis (NEC), a severe emergency for preterm infants, with phages. To this end, the interactions of both culturable and non-culturable phages with the target bacteria will be characterized in vitro and in vivo. Optimized phage cocktails will be tested in clinically relevant pig models. In addition, “*PHARMS*” (duration 2019–2024, funded by an ERC Starting Grant) aims to discover novel antibacterials against *A. baumannii*, *Helicobacter pylori*, and *Haemophilus influenza*, by studying phage–host interactions using culture-independent and multi-omics approaches. “*COVPHA*” (duration 2020–2023, funded by BMBF) aims to identify the co-infecting MDR bacteria in COVID-19 patients using metagenomics and to isolate effective phages against them. In addition, multiple phage endolysins have been identified. The efficacy of the phages and endolysins will be tested in vivo and ultimately applied to patients, under compassionate use and in an early clinical trial. In addition, the Munich Phage Center led by L.D., which is part of the “Center for Integrated Infection Prevention (ZIP)” at the Technical University of Munich (funded by the German Federal Research Building program), aims to develop innovative phage-based therapeutics against MDR pathogens in humans and animals. To this end, a high-throughput phage-culturomics pipeline to automatically isolate and characterize phages as well as a well-characterized phage bank against highly-critical bacteria, including ESKAPE pathogens, are currently being built.

The ‘Therapeutic Phage Group’ of the Bundeswehr Institute of Microbiology, Munich (head J.B.): In collaboration with the European Commission, *Klebsiella* phages and their bacterial hosts are characterized, optimized and prepared for therapeutic applications with a one-health perspective (Joint Programming Initiative on Antimicrobial Resistance, JPIAMR, funded KLEOPATRA consortium, https://www.jpiamr.eu/projects/kleopatra/, accessed on 12 January 2023”; Design and implementation of effective combination of Phages and Antibiotics for improved TheRApy protocols against KLEbsiella pneumoniae). The aim is to expand the host range of phages, optimize cell-free production, support phage therapy in clinical trials and individual healing-trial protocols with personalized phage preparations, and improve therapeutic approaches of clinical partners [20].

### 3.7. One-Health Approach (Food Products, Veterinary Medicine)

‘One-health’ phage applications are increasingly discussed worldwide, but have so far hardly found their way into a holistic approach with respect to the German food chain and veterinary/environmental use. For over a decade, German experts of the different ‘one-health’ compartments and the legislative authorities have been discussing the routine implementation of phages. Major concerns hampering a final decision include esp. (A) a lack of risk assessment for phages released into habitats, (B) the bacterial diversity associated with low reduction and phage-efficiency rates, (C) a spread of phage-resistant bacterial subpopulations associated with a loss of treatment options, and (D) potential disadvantageous effects or changes in microbial communities affecting plant, animal, and human health [31]. Nevertheless, German companies (e.g., PTC Phage Technology Center GmbH) are also invested in the development of preparations for animal/food applications (e.g., against *Salmonella enterica*, *Campylobacter* spp.), but so far, no general permission for the use of phage products exists in Germany. As the clonality and the target bacteria in the veterinary/food sector are broadly overlapping, products for animal use (e.g., livestock) may also be suitable along the food chain. So far, various research projects have confirmed the beneficial effect of different applications, such as bioremediation, biocontrol and bio preservation between animal housing (pre-harvest) to the food product (post-harvest) [32,33,34], but broad safety evaluations including data on the harmful effects on microbial communities, are still lacking. Currently, several institutions, such as the BfR and the Max-Rubner Institute, as well as universities (e.g., TiHo–Department Food Molecular Biology and Antimicrobial Strategies, Hannover) investigate suitable procedures for treatment strategies in/on livestock and food for future application. In these sectors, phage banks and bacterial collections (including *Salmonella*, *Campylobacter*, *Yersinia*, *Klebsiella*, *Vibrio*) have been developed to gain access to a broad spectrum of phages suitable for potential treatment application. In addition, questions addressing the safety of phages are increasingly being studied to close the knowledge gaps for the final risk assessment.

### 3.8. Network Structure and Activities

October 2017: Two symposia, both organized by the University of Hohenheim in Stuttgart, brought the potential of the German phage-researchers’ community to light. The first symposium, in 2017, with 170 participants from 20 countries, addressed the full spectrum of phage research and application. The plenary included experts from the two regulatory bodies BfArM and Paul-Ehrlich-Institut (PEI), as well as from the Federal Ministry of Education and Research (BMBF), who discussed “quo vadis, German phage research”. The regulators stated that there was no clearly defined product to prompt an immediate licensing activity, and urged interested companies to approach the authorities proactively for an early detailed exchange, to avoid costly procedures. They declared their openness to dialogue to support companies planning to develop a phage product. It was also stated that close international cooperation should compensate for the limited number of phage applications worldwide.

December 2019: A workshop within the framework of the NATO HFM-313 Research Task Group (RTG) “Re-introduction of phage therapy in military medicine” (www.sto.nato.int, accessed on 12 January 2023”) was held at the Military Academic Hospital, Berlin. A total of 67 invited participants from 17 nations exchanged views on the topic of the re-introduction of phages into today’s medical space. From individual working groups as well as from the plenary, it was emphasized that the creation of an international register and an easier exchange of already isolated phages would be particularly important to advance phage therapy.

October 2021: Phage therapy was also a topic at the 17th Medical Biodefense Conference in Munich, Germany 2021 (MBDC21). In several sessions (https://military-medicine.com/article/4183-medical-biodefense-conference-2021.html, accessed on 12 January 2023), developments in phage diagnostics and therapy were presented and discussed. The topics of talks included reporter-phage-based detection of bacterial pathogens, phage-receptor-binding proteins for pathogen identification, and an enzyme-linked receptor-binding-protein assay. Further topics included the genetic modification with bactericidal transgenes, non-cellular phage production, issues with GMP production of therapeutic phages in Germany, the efficacy of commercial phage preparations against MRSA, and the possibilities of overcoming the antibiotic resistance in dormant bacteria, as well as the genetic modification of phage TUN1, specific for 4 MRGN *K. pneumoniae*.

May 2022: The second symposium organized by the University of Hohenheim in Stuttgart, with 100 participants from eight countries, was dedicated to temperate phages, their life cycles and roles in ecosystems, molecular structure–function relationships and nanostructures such as DNA transport or cell-specific phage-injection machineries, and virus–microbe–host interactions. Once again, it became clear that more concerted action is needed, as practical application does not keep pace with scientific progress, and regulations need to be adapted to personalized-medicine approaches and other fields of phage application. However, in between the two symposia (2017 and 2022), several externally funded projects emerged and DZIF launched campaign-like activity in the area. In addition, cooperation with the regulators has intensified considerably, as a result.

July 2022: To promote translational phage research, DZIF organized a first strategic meeting on “Bacteriophages in Science and Clinical Use”. The symposium took place in Frankfurt am Main, with 75 national and international participants, including physicians, scientists and industrial partners. Following expert talks on important phage-related research aspects, regulatory requirements and first experiences with phage treatments, all participants actively engaged in further identification of specific needs to promote phage research and clinical application in Germany. As major results of the symposium, a DZIF Translational Phage-Network (short: *DZIF TransPhage-Net*) was founded and a roadmap for translational phage research was drafted.

DZIF has a strong interest in promoting translational phage research, i.e., the successful transfer of research results from bench to bedside. Within the DZIF translational thematic unit “Healthcare-Associated and Antibiotic-Resistant Bacterial Infections” (TTU HAARBI) several initiatives have started up to support phage research in Germany. To provide physicians with an official directive for safe phage therapy, the German Society for Infectious Diseases initiated an official *S2k* guideline for phage therapy, in which medical, pharmacological, and biological societies are currently engaged, including DZIF. In addition to these activities a wide international and interdisciplinary network with a patient-oriented focus, Phage4_1Health Consortium (https://phage4-1health.com/, accessed on 12 January 2023”), originated in Germany in 2020. The combined efforts and expertise of phage scientists, physicians, pharmacists, microbiologists, bioengineers, bioinformaticians, and regulatory affairs officers have joined together to support physicians who intend to treat patients with phages. Topics of the monthly meetings are the presentation of activities of the individual network partners, preparation of funding applications and the identification of suitable partners for clinical and scientific studies. Invited presentations from members and guests on new approaches to phage therapy and the required regulatory processes aim to inform and integrate successful approaches in Europe, Israel, and the US.

A more complete overview of the current phage landscape in Germany is provided through a literature review of recent years and the personal communication of various research groups. The working groups of institutions and companies contributing to translational/clinical phage research are compiled in Table 2.

### 3.9. Most Urgent Changes Needed in the German Phage Landscape, Most Important Research Topics and the Biggest Hurdles for the Implementation of Phage Therapy

Each author provided one or more answers to the three questions presented below in the legends to Figure 2, Figure 3, Figure 4. The frequency of the answers clustered by all 20 authors determined the height of the columns in the corresponding graphs.

As seen in Figure 2, most points were allocated to the production of phages and to a creation of a registry for documentation of all treatment data. The participants expressed the need for standardized protocols for phage therapy and clarification on the topic of healthcare-cost coverage. Randomized controlled trials need to be conducted to prove the efficacy and safety regarding different indications. In parallel, magistral formulations for personalized therapy for indications that are less likely to be covered by conventional approval processes should be developed. Production capacities, according to the relevant regulatory guidelines (such as GMP production for major pathogens with polyvalent phages available and local small-scale production of phages against rare pathogens or those with narrow specificity) need to be established, and parameters for quality control and release determined. To implement phage therapy, dedicated therapy centers have been suggested, and ways to reduce treatment costs must be found. Other points mentioned were: the establishment of either one centralized phage bank or a network of phage banks for clinical use and the continuous isolation of phages, the need for increased funding of applied research, improved networking among research groups and physicians, the raising of awareness of phage therapy among clinicians and the general population, and the need for new methods to rapidly determine the phage susceptibility of clinical isolates.

Research topics that the participants would like to see prioritized are mainly related to optimizing phage therapy and investigating safety aspects (Figure 3). Specifically, research on finding the optimal dose, dosing intervals and duration of therapy, as well as the largely unknown pharmacodynamics and pharmacokinetics were mentioned. Moreover, the need to gain a better understanding of the evolution of phage resistance in the host bacteria and measures to prevent such resistance, as well as of phage–antibiotic synergy, and the impact of phage therapy on biofilms, the human microbiota and immune systems, was expressed. Other important research topics mentioned were the development of preclinical in vivo models and rapid diagnostics for bacteria–phage matching, as well as cell-free production methods for natural, synthetic or genetically engineered phages.

The biggest hurdles towards implementation of phage therapy in Germany, as seen by the participants, include the current lack of approved phage therapeutics, missing uniform guidelines and quality standards, the limited access to phages for clinical use and the lacking infrastructure for phage production, sensitivity testing and application in the clinics (Figure 4). Moreover, the lack of awareness of phage therapy, including its limitations, among physicians and the public was mentioned.

## 4. Discussion

The aim of this article was to present the current German-phage-therapy landscape from the perspective of clinicians, researchers working closely with them, and those responsible for regulatory aspects, i.e., a group representing the present efforts of physicians, pharmacists, microbiologists, bioengineers, bioinformaticians and regulators.

As phages are currently not clinically approved, phage therapy is only being applied to a limited number of patients in university hospitals and a German armed-forces hospital, which operate without exception under the umbrella of Article 37 of the Declaration of Helsinki. Either ready-to-use composite phage products are used (typically procured from abroad) or phages are produced within the framework of permission-free production in the respective hospital for their own patients, under supervision of the treating physician. To date, there is only one place in Germany that can produce phages on a larger scale, according to GMP (Fraunhofer ITEM in Braunschweig). These GMP phages will initially benefit the publicly funded research projects *Phage4Cure* and *PhagoFlow*, mainly with phages against *P. aeruginosa* and *S. aureus*. Due to the current and previous limitations, there are probably hardly 100 patients who have been treated in Germany in the immediate past, in marked contrast to the many thousands of phage applications between the 1930s and the 1960s. Nevertheless, the treatment volume will increase from 2023 onwards, due to the imminent start of clinical phases of the ongoing research projects. It is also becoming apparent that numerous clinics will take advantage of permission-free production of medicinal products (§13 (2b) German Medicinal Products Act) in the foreseeable future, to produce phages for their patients.

It is striking that different phage banks exist in Germany, but, due to their different institutional backgrounds and the policies to which they are bound (e.g., bioresource center or university based), they do not operate in comparable modes, and do not freely exchange all their phage holdings. There is also no full genome sequencing of the respective complete phage collections. Thus, a complete and common data collection of all bioinformatic data and the respective host areas is not yet available. However, the DSMZ envisages sequencing the genomes of all clinically relevant phages quickly.

A major limitation of the present analysis is certainly the arbitrary selection of authors, which cannot cover all the important groups contributing to phage therapy and research in Germany. However, the group encompasses stakeholders from most—if not all—relevant German phage-research consortia, so that overall a valid portrait of the situation in Germany could be generated. In addition, there are numerous research groups engaged in translational research, for example, to facilitate the bacteria–phage matching process [47] or exploit phage–antibiotic synergy [50], amongst others (Table 2), although some of these groups are only involved in providing specific analytical methods. Moreover, a priority program funded by DFG initiated in 2021—*SPP2330*—aims to elucidate the regulation of the phage life cycle, identify new bacterial anti-phage defense systems, and study the impact of phages on viral communities and biofilms (https://spp2330.de/, accessed on 12 January 2023). These research efforts will also help to further improve the efficacy of phage therapy.

Therefore, although the funding situation and awareness require further improvement, important steps have been taken en route to addressing the antibiotic-resistance challenge with phage therapy. It must also be emphasized that large sums of public funding are already being used for research projects, and that the expert report on phage therapy commissioned by the German Bundestag in 2021—expected to be published in 2023—also underlines the serious interest of political decision makers. In the opinion of the authors, following the demonstration of the clinical efficacy of phage therapeutics, the current legal framework for the use and production of phages (currently preventing widespread clinical use) should be adapted to the special requirements of phages, to ensure patient access in a timely manner. However, the present regulatory situation can also be viewed positively. The current relatively low level of information in the population and the legal situation mostly prevents uncritical use of ill-defined phage preparations. The broad application of phage therapy without appropriate rules and scientific exchange of treatment results would probably result in unnecessary failures, and may contribute to discrediting the therapy. The worst case would be that phage therapy would be abandoned again, before proper implementation—a situation resembling the first half of the 1940s in the US and Germany. Thus, establishing controlled, high-quality phage production and therapy is paramount.

Against the background of these considerations, the German regulatory authorities support phage therapy and the respective approaches for patient access, i.e., a pragmatic approach of magistral production for selected research, and clinical facilities in Germany. Options such as the Belgian model, i.e., a legal framework that allows for individualized phage therapy using magistral formulations (see reference [51] for details) should be considered. In parallel, the pharmaceutical legislation needs to be amended to facilitate phage therapy. Harmonized quality standards should be implemented via the Ph. Eur. general chapter on PTMPs.

A nation- or EU-wide common approach to modern phage therapy, developed together with international actors, is the ideal. Indeed, institutional dependencies and personally motivated efforts to commercialize phage production and therapy may lead to demarcations and limited exchange of information. Nevertheless, the current situation shows activity supported by great enthusiasm in investigating the potential and implementing phage therapy in Germany as quickly as possible. So far, the stakeholders have been substantially supported by the regulatory authorities. At present, there is an urgent need to have more phages available for clinical testing and use, to expand production capacities, develop quality standards for production and clinical use, establish a registry with structured documentation of all treated patients, and gain more scientific evidence for the effectiveness of phage therapy. Current important scientific questions should urgently be answered through targeted networking and funding of translationally oriented research institutions, so that the clinical use of phage products in Germany can take a crucial step forward in the fight against multi-resistant bacteria.

## Figures and Tables

**Figure 1 viruses-15-00588-f001:**
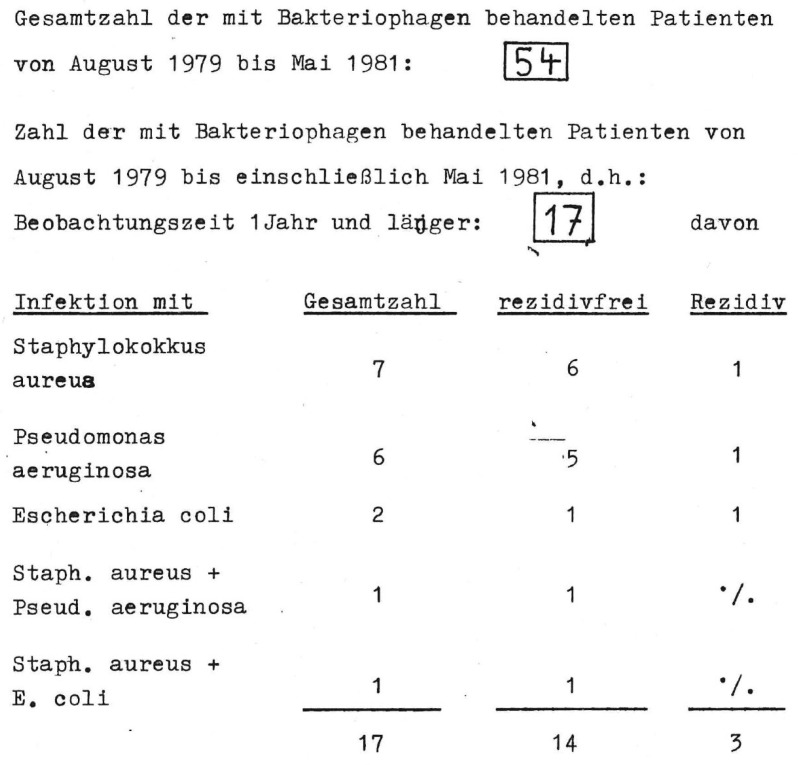
Scan of the original records of H. Lodenkämper, orthopedic surgeon at the Endo-Klinik in Hamburg, Germany. Out of the 54 patients with an implant-associated infection (“*Gesamtzahl*”) only 17 were followed up for 1 year or longer. Of these, 14 (~82%) were free of recurrence (“*rezidivfrei*”).

**Figure 2 viruses-15-00588-f002:**
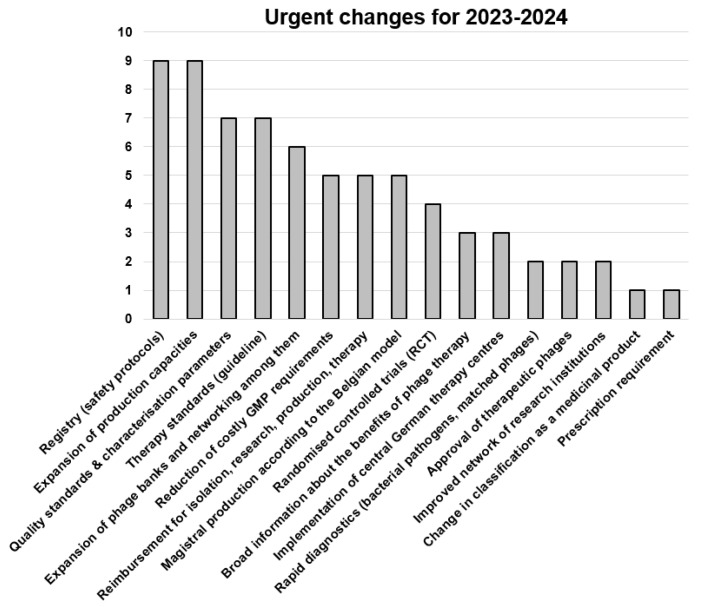
Responses to the question: “What changes (in any field) do you think are urgent (1–2 years) in order to implement phage therapy in Germany as soon as possible in the solidarity community (e.g., nationwide, more than 1000 treatments per year)?” Data are based on the answers of the 20 authors of this article, with 3.6 answers per participant, on average. RCT, randomized controlled trial; ITT, individual-treatment trial.

**Figure 3 viruses-15-00588-f003:**
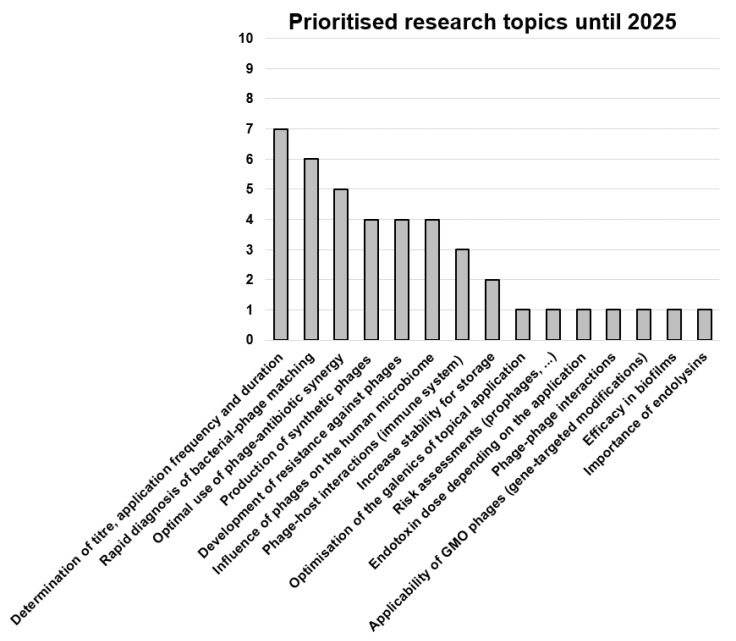
Responses to the question: “Which research topics should be addressed now, prioritized immediately (time frame until 2025), in order to be able to use an effective phage therapy as soon as possible?” Data are based on the answers of the 20 authors of this article, with 2.1 answers per participant, on average. GMO, genetically modified organism.

**Figure 4 viruses-15-00588-f004:**
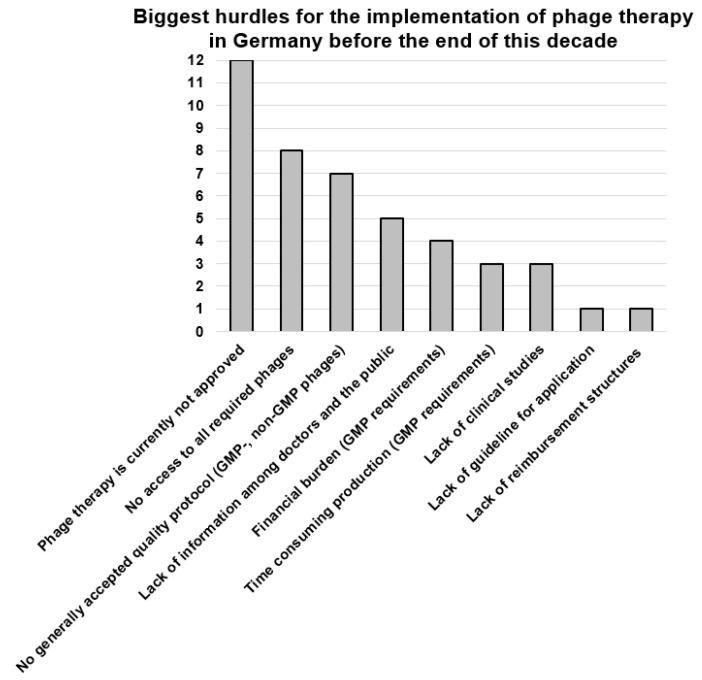
Responses to the question: “What are the biggest hurdles today for the implementation of phage therapy in Germany before the end of this decade?” Data are based on the answers of the 20 authors of this article, with 2.2 answers per participant, on average. GMP, good manufacturing practice.

**Table 1 viruses-15-00588-t001:** Legal frameworks for the use of phages in Europe. API, active pharmaceutical ingredient; FDA, Food and Drug Administration; GMP, good manufacturing practice.

Compassionate-Use Program (CUP)	“Individual-Treatment Trial”	Magistral Preparation
“Compassionate use” is usually included in the respective national medicinal-products act. In Europe, the European Medicines Agency (EMA) provides recommendations through the Committee for Medicinal Products for Human Use (CHMP), but these do not create a legal framework. CUP are coordinated and implemented by Member States, which set their own rules and procedures (Article 83 of Regulation (EC) No 726/2004). ◦ Is granted for a group of (unnamed) patients ◦ Life-threatening, long-lasting, or severely debilitating conditions that cannot be satisfactorily treated with any currently approved drug ◦ Realistic probability of therapeutic benefit ◦ Patients should always be considered for inclusion in clinical trials before being offered CUP. ◦ Prior approval from the competent authority (e.g., EMA, FDA) is required. Medicinal product is included in an official list of currently running indicated-hardship programs. ◦ Already in a centralized marketing-authorization procedure or under clinical investigation (e.g., Germany: at least phase II clinical trial; USA: phase I clinical trial; Switzerland: no trial required). ◦ Product is manufactured according to the principles and guidelines of GMP. ◦ Inclusion and exclusion criteria must be stated, single patient or small-group use. ◦ Provided free of charge for use in patients in some countries.	“Individual treatment trials” are ethically based on Article 37 of the Helsinki Declaration: “In the treatment of an individual patient, where proven interventions do not exist or other known interventions have been ineffective, the physician, after seeking expert advice, with informed consent from the patient or a legally authorized representative, may use an unproven intervention if in the physician’s judgement it offers hope of saving life, re-establishing health or alleviating suffering. This intervention should subsequently be made the object of research, designed to evaluate its safety and efficacy. In all cases, new information must be recorded and, where appropriate, made publicly available.” * ◦ Therapeutic trial always relates only to a specific/individual patient “*Named Patient Use*” ◦ Within the scope of the physician’s freedom of therapy, with the consent of the patient ◦ Life-threatening, long-lasting, or severely debilitating conditions that cannot be satisfactorily treated with any currently approved drug. ◦ No approval is required. ◦ Under the direct responsibility of the physician. ◦ Significantly higher standard of care in the physician’s approach (risk–benefit ratio, education). ◦ Case-by-case decision. ◦ Healing the individual patient is priority. ◦ No gain of knowledge in the sense of a research study. ◦ Drug/active ingredient, therapy plan, dosage, occurrence of undesirable effects and the general course of treatment are described in detail.	In European law, notion of a magistral preparation (compounded-prescription-drug product in the US) is defined as “any medicinal product prepared in a pharmacy in accordance with a medical prescription for an individual patient” (Article 3 of Directive 2001/83 and Article 6 quarter). Today, there are no formal guidelines regarding the clinical use (e.g., medical indications, formulations, and posology) of magistral phage drugs. ◦ Pragmatic way to personalize patient treatments to specific needs and to make medications available that do not exist commercially. ◦ For a given individual, named patient, according to a prescription by a physician. ◦ Following the technical and scientific standards of the pharmaceutical art. ◦ APIs of magistral preparations must meet the requirements of the European Pharmacopoeia or of the national Pharmacopoeia. The relevant properties and qualities of the API should be defined in an internal monograph (prepared by the supplier). ◦ A certified laboratory performs external quality testing to evaluate the properties and quality of the API. ◦ For phage activity and bacterial-susceptibility testing, a “phagogram” is performed ◦ The phage API is then submitted to the hospital pharmacy for incorporation into magistral formulations. ◦ Magistral formulations should be delivered under the direct responsibility of a physician.

* The declaration is not a legal framework and cannot be legally enforced.

**Table 2 viruses-15-00588-t002:** Working groups (non-exhaustive) of German research institutions and companies contributing to translational/clinical phage research, based on a PubMed search of the years 2020–2022 (as of 31 August 2022) and personal communication with research groups. The working groups of the authors of this article are not included. * In the case where there is no publication yet, not applicable (n.a.).

Institution or Company	City	Non-Exhaustive Research Topics (Year of First Publication) *
Anhalt University of Applied Science, Applied Biosciences and Process Engineering (https://www.hs-anhalt.de/hochschule-anhalt/angewandte-biowissenschaften-und-prozesstechnik/uebersicht.html)	Köthen	Bacterium-phage matching, metaproteomics (2019 [35])
BacTrace BioTec AG (https://www.bactrace.de/en/home-english/)	Munich	Development of in-vitro-evolved phages (2021 [36])
Berlin University of Applied Sciences, School of Life Sciences and Technology, Department of Microbiology (https://www.bht-berlin.de/labor/detail/mib)	Berlin	Microbiome dynamics exposed to phages, interplay between phages and antimicrobials applied on biofilms, phage–antibiotic synergy (n.a.)
Carl von Ossietzky University Oldenburg, Institute for Chemistry and Biology of the Marine Environment (http://moraru-phage-lab.icbm.de/)	Oldenburg	Phage–antibiotic antagonism (2022 [37])
Charité—Universitätsmedizin Berlin, Center for Musculoskeletal Surgery and Berlin Institute of Health (https://cmsc.charite.de/ and https://www.bihealth.org/)	Berlin	Phage therapy designed for biofilm infections (2018 [38])
Coburg University of Applied Sciences, Institute of Bioanalysis (https://bioanalytik.co/en/home)	Coburg	Application of therapeutic peptides on recombinant phages (n.a.)
FINK TEC GmbH (https://www.finktec.com/)	Hamm	Use of phage cocktail in veterinary medicine and food technology (2020 [39])
Forschungszentrum Jülich, Institute of Bio- und Geosciences (https://www.fz-juelich.de/en/ibg)	Jülich	Phage–antibiotic antagonism (2022 [37])
Fraunhofer Institute for Interfacial Engineering and Biotechnology (https://www.igb.fraunhofer.de/en.html)	Stuttgart	Implementation of selected phages in suitable formulations (n.a.)
Free University (FU) Berlin, Institute of Chemistry and Biochemistry (https://www.fu-berlin.de/en/einrichtungen/fachbereiche/fb/bio-chem-pharm/chm/index.html)	Berlin	Interaction of phage extracellular glycosidases with bacterial biofilm, biofilm penetration by phages (n.a.)
Friedrich-Schiller University, Institute of Physical Chemistry (https://www.ipc.uni-jena.de/en)	Jena	Development of in-vitro-evolved phages (2021 [36])
GEOMAR Helmholtz Centre for Ocean Research (https://www.geomar.de/en/)	Kiel	Imaging approach to studying phage distribution and cellular association (2021 [40])
Heidelberg University, Institute for Molecular Systems Engineering and Advanced Materials (https://www.imseam.uni-heidelberg.de/)	Heidelberg	Platform technology based on genetically modified phages (2019 [41])
Helmholtz-Centre for Infection Research (HZI), Central Facility for Microscopy (https://www.helmholtz-hzi.de/en/research/technology-platforms/overview/zeim/our-expertise/)	Braunschweig	Phage cocktails for veterinary medicine and food technology (2020 [39])
InfectoGnostics Research Campus (https://www.infectognostics.de/)	Jena	Development of in-vitro-evolved phages (2021 [36])
Justus-Liebig University Giessen, Clinic for Urology, Pediatric Urology and Andrology (https://www.ukgm.de/ugm_2/deu/ugi_uro/index.html)	Giessen	Development of in-vitro-evolved phages (2021 [36])
Leibniz Institute for Photonic Technologies Jena e.V. (https://www.leibniz-ipht.de/en/homepage/)	Jena	On-site identification of phage-mediated bacterial lysis (n.a.); development of in-vitro-evolved phages (2021 [36])
Leibniz Research Institute for Molecular Pharmacology (https://leibniz-fmp.de/)	Berlin	Development of biofilm-penetrating recombinant phages (n.a.)
Max Planck Institute for Developmental Biology, Department of Microbiome Science (https://www.bio.mpg.de/48843/microbiome-science-ruth-ley)	Tübingen	Computational tools for the analysis of uncultivated phage genomes (2019 [42])
Max Planck Institute for Evolutionary Biology (https://www.evolbio.mpg.de/2169/en)	Plön	Phage resistance affected by antibiotics (2022 [43])
Max Planck Institute for Medical Research, Department for Cellular Biophysics (https://www.mr.mpg.de/13943505/cellular_biophysics)	Heidelberg	Synthetic phages for personalized treatment on demand (n.a.)
Max Planck Institute for Intelligent Systems	Stuttgart	Genetically engineered phages (2019 [41])
Max Planck Institute for Molecular Genetics, Department Computational Molecular Biology (https://www.molgen.mpg.de/en/bioinformatics)	Berlin	Machine-learning framework for translatable phage research (n.a.)
Max Planck Institute for Terrestrial Microbiology (https://www.mpi-marburg.mpg.de/)	Marburg	Coexistence of phage-resistant and phage-susceptible bacteria (2020 [44])
Max Planck Institute of Colloids and Interfaces, Department Theory and Bio-Systems (https://www.mpikg.mpg.de/theory)	Potsdam	Interaction of extracellular glycosidases of phages with bacterial biofilm (2021 [45])
Max Rubner Institute, Institute of Microbiology and Biotechnology (https://www.mri.bund.de/en/institutes/microbiology-and-biotechnology/)	Kiel	Developing of a broad-spectrum phage collection and phage-mediated manipulation of the gut microbiome (2020 [46])
Max Delbruck Center for Molecular Medicine, Crystallography (https://www.mdc-berlin.de/heinemann)	Berlin	Interaction of extracellular glycosidases of phages with bacterial biofilm (2021 [45])
Medea Biopharma (https://www.medea-bio.com/)	Munich	Development of a scalable phage-therapy pipeline (n.a.)
Paul Ehrlich Institute (https://www.pei.de/EN/home/home-node.html)	Langen	Kinetic fingerprinting to infer phage–host interactions (2020 [47])
Philipps-University Marburg, Department of Physics (https://www.uni-marburg.de/en/fb13)	Marburg	Coexistence of phage-resistant and phage-susceptible bacteria (2020 [44])
PTC Phage Technology Center GmbH (https://www.finktec.com/applied-phage)	Bönen	Phages in veterinary medicine and food technology, phages against uropathogenic bacteria (n.a.)
Robert Koch Institute (https://www.rki.de/EN/Home/homepage_node.html)	Berlin	Phage communities over evolutionary history (2021 [48])
Robert Koch Institute (https://www.rki.de/EN/Home/homepage_node.html)	Wernigerode	Use of phage cocktail in veterinary medicine and food technology (2020 [39])
RWTH Aachen University, Institute of Biotechnology (https://www.biotec.rwth-aachen.de/go/id/imne/?lidx=1)	Aachen	Phage–antibiotic antagonism (2022 [37])
RWTH Aachen University, Institute of Medical Microbiology (https://www.medizin.rwth-aachen.de/cms/Medizin/Die-Fakultaet/Institute-und-Kliniken/Die-Institute/Klinisch-theoretische-Institute/~ezkx/Institut-fuer-Medizinische-Mikrobiologie/)	Aachen	Phage–antibiotic synergy (2018 [49])
Technical University Dresden, Institute of Medical Microbiology and Hygiene (https://tu-dresden.de/med/mf/mib)	Dresden	Development of in-vitro-evolved phages (2021 [36])
University Hospital Münster, Institute of Hygiene (https://www.ukm.de/institute/hygiene)	Münster	Phages against uropathogenic bacteria (n.a.)
University of Hamburg, Institute of Biochemistry and Molecular Biology (https://www.chemie.uni-hamburg.de/en/institute/bc.html)	Hamburg	Synergistic/antagonistic interactions of bacterium, phage, and antibiotic (n.a.)
University of Potsdam, Physical Biochemistry Group (https://www.uni-potsdam.de/en/ibb-physbiochem/index)	Potsdam	Interaction of phage extracellular glycosidases with bacterial biofilm (2021 [45])
University of Stuttgart, Institute for Materials Science (https://www.imw.uni-stuttgart.de/en/)	Stuttgart	Genetically engineered phages (n.a.)
University of Tubingen, Interfaculty Institute for Microbiology and Infection Medicine (https://uni-tuebingen.de/en/faculties/faculty-of-science/departments/interfaculty-facilities/imit/)	Tübingen	Predicting the host range of phages, based on computational clustering (n.a.)
University of Veterinary Medicine Hannover, Institute for Food Quality and Food Safety (www.tiho-hannover.de)	Hannover	Phage cocktails for veterinary medicine and food technology (2020 [39])
University of Wurzburg, Imaging Core Facility (https://www.biozentrum.uni-wuerzburg.de/em/startseite/)	Würzburg	Imaging approach to studying phage distribution and cellular association (2021 [40])
Zuse Institute Berlin (https://www.zib.de/)	Berlin	Mathematical methods for predicting bacteria–phage interactions and possible side effects (n.a.)

All websites of the table have been accessed on 2 February 2023.

## Data Availability

Not applicable.

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
