# Peer review of "Phage Therapy in Germany—Update 2023"

_viruses, 2023, doi:10.3390/v15020588_

Round 1

Reviewer 1 Report

This manuscript in the form of a communication summarises the current state of phage therapy in Germany. The article is well written and gives a comprehensive coverage of phage therapy projects, phage banks and importantly the regulatory situation it stands in early 2023. A strength of the article lies in the authors being drawn from a network of participating institutions across Germany and the inclusion of a series of suggestions from the group regarding key changes required to drive the development of phage therapy. The historical overview was fascinating and a clear link was made to the need to proceed cautiously and rigorously as phage therapy is revived.  Is there any data on the effectiveness of the large scale prophylaxis in the 1960’s (line 204?).

The summary of the legal framework for phage therapy in Germany will be of particular interest to phage therapy proponents based in other countries.

No major concerns.

Minor suggestions/comments. Is there a way to provide references/websites for the summary of projects presented in Table 2?

Line 337: “The above mentioned production process…” is a little obscure given there are two different approaches mentioned in section 3.3. Which process is being referred to?

The authors refer a couple of time to the Belgian model of phage therapy (line 68; line 791). It would be helpful to give a brief description of exactly what the Belgian approach involves.

Author Response

This manuscript in the form of a communication summarises the current state of phage therapy in Germany. The article is well written and gives a comprehensive coverage of phage therapy projects, phage banks and importantly the regulatory situation it stands in early 2023. A strength of the article lies in the authors being drawn from a network of participating institutions across Germany and the inclusion of a series of suggestions from the group regarding key changes required to drive the development of phage therapy. The historical overview was fascinating and a clear link was made to the need to proceed cautiously and rigorously as phage therapy is revived.  Is there any data on the effectiveness of the large scale prophylaxis in the 1960’s (line 204?).

Response: There is no efficacy data, but the prophylactic measure was discontinued in June 1962, which likely means that the epidemic was considered overcome. This information has been added to the manuscript.

The summary of the legal framework for phage therapy in Germany will be of particular interest to phage therapy proponents based in other countries.

No major concerns.

Minor suggestions/comments. Is there a way to provide references/websites for the summary of projects presented in Table 2?

Response: As suggested, we now provide the websites and references.

Line 337: “The above mentioned production process…” is a little obscure given there are two different approaches mentioned in section 3.3. Which process is being referred to?

Response: Thank you - the sentence has been clarified.

The authors refer a couple of time to the Belgian model of phage therapy (line 68; line 791). It would be helpful to give a brief description of exactly what the Belgian approach involves.

Response: We have included a brief description and a reference to the Belgian approach in the Discussion section of the revised manuscript.

Thank you very much for your effort to help us improve our manuscript!

Reviewer 2 Report

The author's communication on "Phage therapy in Germany-Update 2023" is a very important report and it is well structured to include the status of phage research in Germany and future plans. 

Some minor questions:

Lines 175-177: What is the status of clinical trials in Germany?

Lines 230-232: (and throughout the manuscript) It is better to add references if published or else add a statement that it hasn't been published separately. [As we know previously, the unpublished phage therapy data goes unrecognized]

Lines 242-248: (and throughout the manuscript) What is the outcome of therapy?

Line 273: Intesti phage cocktail is used against?

Lines 332-334: Any advantage of using personalised phage therapy in that case?

Table 2: Is it possible to add a column to include years of research? For example, a company/institute is working on this project/topic since 1988. (Considering phage therapy as a renewed in many parts of the world)

Author Response

The author's communication on "Phage therapy in Germany-Update 2023" is a very important report and it is well structured to include the status of phage research in Germany and future plans.

Some minor questions:

Lines 175-177: What is the status of clinical trials in Germany?

Response: The current state of clinical trials is already covered in Section 3.6, to which we now refer at the end of Section 3.1 of the revised manuscript. For improved clarity the title of Section 3.6 now also includes the term “clinical trials”.

Lines 230-232: (and throughout the manuscript) It is better to add references if published or else add a statement that it hasn't been published separately. [As we know previously, the unpublished phage therapy data goes unrecognized]

Response: Wherever appropriate we either added an additional reference or state that data hasn’t been published separately.

Lines 242-248: (and throughout the manuscript) What is the outcome of therapy?

Response: Thank you for pointing this out. We now describe the outcome of the therapy here and also throughout the manuscript, if data is available. For some historical data, however, there is no efficacy data available, which we now explicitly mention where appropriate.

Line 273: Intesti phage cocktail is used against?

Response: We added this information.

Lines 332-334: Any advantage of using personalised phage therapy in that case?

Response: Klebsiella phages are typically highly strain-specific since for most of them, the primary cellular receptor is the highly diverse capsular polysaccharide. We have added this information to the revised manuscript.

Table 2: Is it possible to add a column to include years of research? For example, a company/institute is working on this project/topic since 1988. (Considering phage therapy as a renewed in many parts of the world)

Response: We have included references and years of research to Table 2.

Thank you very much for your effort to help us improve our manuscript!

Reviewer 3 Report

“Phage Therapy in Germany-Update 2023”

Authors, as interdisciplinary group phage research, in Germany report past, present, and future of phage therapy in Germany as communication for the journal. The regulatory matters for phage therapy, history of phage therapy in Germany, and present phage therapy situation elsewhere would be very interesting. Phages and bacteria as resources for therapy, the bank maintenance and standard of procedures would be very important for the sustainable translational researches. Research activities in recent days would stimulate other related scientists and regulatory scientists for the right direction of the phage therapy in other countries. The results of the survey are also very interesting and I think they will be very helpful.

Overall, the context is as long as the form of communication, but for relevant readers this report will be very interesting and valuable.

Minor, I recommend some English proof reading before last submission.

Author Response

Authors, as interdisciplinary group phage research, in Germany report past, present, and future of phage therapy in Germany as communication for the journal. The regulatory matters for phage therapy, history of phage therapy in Germany, and present phage therapy situation elsewhere would be very interesting. Phages and bacteria as resources for therapy, the bank maintenance and standard of procedures would be very important for the sustainable translational researches. Research activities in recent days would stimulate other related scientists and regulatory scientists for the right direction of the phage therapy in other countries. The results of the survey are also very interesting and I think they will be very helpful.

Overall, the context is as long as the form of communication, but for relevant readers this report will be very interesting and valuable.

Minor, I recommend some English proof reading before last submission.

Response: We carefully corrected the manuscript for language.

Thank you very much for your effort to help us improve our manuscript!